# Cosmology of the companion-axion model: dark matter, gravitational waves, and primordial black holes

Zhe Chen[1], Archil Kobakhidze[1], Ciaran O'Hare[1], Zachary S. C. Picker[1,2]*, and Giovanni Pierobon[3]

**1** School of Physics, The University of Sydney and ARC Centre of Excellence for Dark Matter Particle Physics, NSW 2006, Australia
**2** Department of Physics and Astronomy, University of California, Los Angeles, Los Angeles, California, 90095-1547, USA
**3** School of Physics, The University of New South Wales, Sydney NSW 2052, Australia
* zpicker@physics.ucla.edu

## Abstract

**The companion-axion model introduces a second QCD axion to rescue the Peccei-Quinn solution to the strong-CP problem from the effects of colored gravitational instantons. As in single-axion models, the two axions predicted by the companion-axion model are attractive candidates for dark matter. The model is defined by two free parameters, the scales of the two axions' symmetry breaking, so we can classify production scenarios in terms of the relative sizes of these two scales with respect to the scale of inflation. We study the cosmological production of companion-axion dark matter in order to predict windows of preferred axion masses, and calculate the relative abundances of the two particles. Additionally, we show that the presence of a second axion solves the cosmological domain wall problem automatically in the scenarios in which one or both of the axions are post-inflationary. We also suggest unique cosmological signatures of the companion-axion model, such as the production of a $\sim 10$ nHz gravitational wave background, and $\sim 100\, M_\odot$ primordial black holes.**

# 1   Introduction

The axion is a pseudo-Nambu-Goldstone boson that emerges in theories incorporating the Peccei-Quinn (PQ) solution to the strong CP problem [1–8]. Interestingly, phenomenologically viable axions can also comprise some or all of the observed dark matter (DM) [9–13]. In recent articles [14,15], we argued that a genuine solution to the strong CP problem may require an additional 'companion' axion—the key reason being that additional CP-violation is induced in the Standard Model by charged gravitational instantons, thereby invalidating the original PQ solution [16].

This issue can be resolved by extending the original $U(1)_{\mathrm{PQ}}$ symmetry to $U(1)_{\mathrm{PQ}} \times U(1)'_{\mathrm{PQ}}$. Both of the $U(1)$ subgroups carry mixed $U(1)$-QCD and gravitational anomalies and are spontaneously broken at scales $f_a$ and $f'_a$ respectively. Consequently, two pseudo-Goldstone axions are predicted in the low-energy spectrum of the theory.

Just like the standard QCD axion, the two companion axions should also be considered viable DM candidates. In this article, we study their production in the early universe by classifying several different cosmological scenarios in analogy to past literature on single-axion cosmology (see e.g. [17–22] and references therein). In principle, one could envisage three distinct scenarios: (i) both PQ phase transitions occur before/during inflation and hence both axions are produced through the misalignment mechanism [9–11] with two unknown but fixed initial angles; (ii) the lighter axion is produced before inflation, while the second PQ transition occurs after inflation and the corresponding heavier axion is produced with a predictable distribution of angles; (iii) both axions are produced after inflation (or inflation never occurs), leading to a single predictable DM abundance from misalignment, but a potentially complicated network of topological defects.

The misalignment mechanism for the companion axion model is more involved than for conventional models, with potentially richer ensuing dynamics. Another rather generic implication is that any axion domain walls produced in post-inflationary scenarios are automatically unstable—removing the need for *ad hoc* PQ symmetry breaking terms. As we will show, the dynamics of these unstable walls in the early universe results in the production of gravitational waves (GWs) (see also [23–28]) and primordial black holes (see also [29, 30]), potentially providing further signatures of the companion axion model, complementary to laboratory searches [15].

# 2   Companion-Axion model

We consider a $U(1)_{\mathrm{PQ}} \times U(1)'_{\mathrm{PQ}}$ extension of the original PQ symmetry. Once spontaneously broken, this results in *two* pseudo-Goldstone axions with the following potential:

$$
\begin{aligned}
V(a, a') = - &K \cos \left( N \frac{a}{f_a} + N' \frac{a}{f'_a} + \theta \right) \\
- &\kappa K \cos \left( N_g \frac{a}{f_a} + N'_g \frac{a}{f'_a} + \theta_g \right) ,
\end{aligned}
\tag{1}
$$

where $K \simeq (75.6 \, \mathrm{MeV})^4$ [36] and $\kappa \sim 0.04$–$0.6$ [14]. The lowest energy state is realized for the axion field expectation values that cancel out both CP-violating terms, leading to the constraint $N N'_g \neq N' N_g$ [15]. The two axion states $a$ and $a'$ are mixed through the interactions in the potential and the corresponding mass eigenstates are,

$$
\begin{aligned}
a_1 &= a \cos \alpha - a' \sin \alpha , \\
a_2 &= a \sin \alpha + a' \cos \alpha ,
\end{aligned}
\tag{2}
$$

where $\alpha$ is the mixing angle. Following [15], we will often refer to two regimes: a hierarchical regime, $f_a \ll f'_a$; and a strong mixing regime, $f_a \approx f'_a$. The $a_1$ axion reproduces the standard QCD axion, up to $\mathcal{O}(1)$ differences related to the anomaly coefficients, while the second axion's

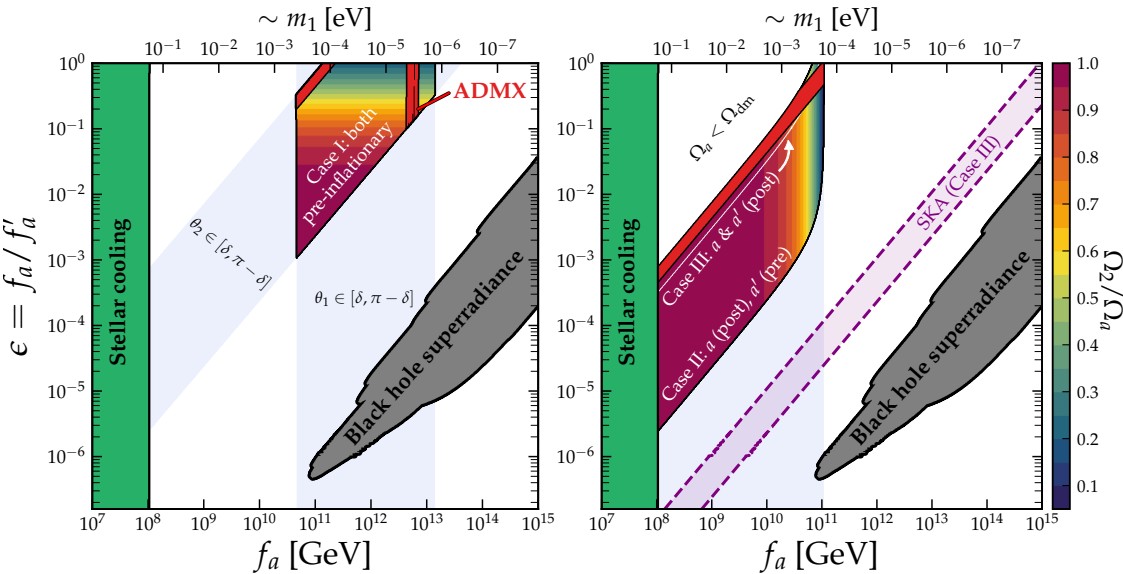

Figure 1: The companion axion parameter space $(f_a, \epsilon)$, showing the range of parameters for which the two axions represent viable DM candidates. The two panels show the preferred regions for our three cosmological scenarios, where the color scale encodes the fraction of DM comprised of the lighter axion ($a_2$). Case I (left panel, colored region) is when both axions are pre-inflationary. Case II (right-panel, colored region) is when just $f_a$ is post-inflationary, and Case III (right-panel, white line) is when both axions are post-inflationary. Assuming the two axions couple to the photon, then the region colored in red is excluded by ADMX [31–35] (see [15] for more details). In Case III specifically we can also predict a region of parameter space (shown in purple) when $m_{1,2} \sim 10^{-9}$ eV which could be probed in the future by SKA via the stochastic gravitational wave background generated by the collapsing domain walls.

mass can be significantly lighter:

$$m_1^2 \sim 2K/f_a^2, \qquad\qquad m_2^2 \sim \kappa\epsilon^2 m_1^2, \qquad\qquad (3)$$

where $\epsilon \equiv f_a/f_a'$. For concreteness, in our numerical results we adopt $\{N, N', N_g, N_g'\} = \{3, 1/2, 13/2, 3/2\}$ and $\kappa = 0.04$. Alternative choices result only in minor quantitative differences.

## 3  Dark matter

We begin by estimating the relative DM abundances in the two axions generated via the misalignment mechanism. We ignore for the moment axion production due to the string-domain wall network that forms in the post-inflation PQ breaking scenario, since this already demands a full numerical treatment in the single-axion case (see for example [37–44] for recent activity in this area). There are three possible scenarios for companion-axion misalignment:

(I)  Both PQ symmetries are broken before the end of inflation (i.e. both axions are "pre-inflationary")

(II)  The $a'$ symmetry is broken before the end of inflation, while $a$ is post-inflationary.

(III)  Both axions are post-inflationary[1].

The basic idea of the misalignment mechanism involves each of the axion fields rolling down

---

[1]This case also trivially corresponds to non-inflationary cosmology.

their shared potential from some initial values misaligned from 0 by some $\theta_{1,2} \in [-\pi, \pi]$. When $m_{1,2}(t) \gtrsim 3H(t)$, the fields begin to oscillate around the CP-conserving minimum. The energy density stored in the zero-modes of the axion fields can then be interpreted as cold DM, with abundances proportional to the square of those initial angles.

We will always attempt to satisfy the restriction that the abundance in the two axions neither exceeds nor falls short of the observed DM abundance $\Omega_{\mathrm{dm}}h^2 = 0.12$ [45]. As in the single-axion model, this will mean that only certain values of $\theta_{1,2}$ result in the correct abundance, but not all of the scenarios offer the freedom to choose $\theta_{1,2}$. Rather, the possible values depend upon the ordering of the various cosmological epochs:

(I) Both $\theta_{1,2}$ could plausibly take any value from $-\pi$ to $\pi$, and the field would take on a single value within the horizon after inflation.

(II) $\theta_2$ can take any single value, but $\theta_1$ takes on different values in different causal patches, leading to an ensemble of values all entering the horizon as the Universe expands. The relevant angle to use when calculating the abundance is then the stochastic average $\theta_1 \equiv \sqrt{\langle \theta^2 \rangle} = \pi/\sqrt{3}$.

(III) We use the average $\theta_{1,2} = \pi/\sqrt{3}$ for both angles.

Usually an anthropic argument can be made in the pre-inflationary case for the angle to be tuned arbitrarily close to $\pm\pi$ to maximize the DM abundance, or towards 0 to limit it. If we prefer to avoid any such fine-tuning we can introduce a small parameter $\delta$ and define a "natural" pre-inflationary window for angles $|\theta_{1,2}| \in [\delta, \pi - \delta]$. We will take $\delta = 0.1$ for display purposes in Fig. 1, but the cutoff is arbitrary.

The abundance of DM can be found from the zero-mode evolution of the two axion fields, described by a linearized system of coupled oscillators:

$$\partial_t^2 a + \frac{3}{2t}\partial_t a + M_{11}a + M_{12}a' = 0\,,$$
$$\partial_t^2 a' + \frac{3}{2t}\partial_t a' + M_{22}a' + M_{21}a = 0\,, \tag{4}$$

where $M_{ij}$ are elements of the thermally corrected axion mass matrix, and the Hubble damping term is evaluated for a radiation dominated Universe, $H = 1/2t$. For simplicity, we work in the linear (harmonic) approximation and ignore non-linear terms present in the full potential [18, 46–49].[2]

When $\epsilon \ll 1$, we expect the lighter axion to substantially dominate the final abundance due to the hierarchy $f_a' \gg f_a$. The mass matrix in this limit is,

$$M = m_1^2(T)\begin{pmatrix} 1 & -\epsilon^2 \\ -\epsilon^2 & \kappa\epsilon^2 \end{pmatrix} + \mathcal{O}(\epsilon^4) \tag{5}$$

where for the heavier mass we have adopted the standard thermal axion mass calculation from [19],

$$m_1^2(T) = \min\left[m_1^2, m_1^2\left(\frac{\widetilde{T}}{T}\right)^n\right]\,, \tag{6}$$

with $n = 6.68$ and $\widetilde{T} = 103\,\mathrm{MeV}$ [19]. We obtained the thermal mass matrix Eq.(5) by simply elaborating on the calculation of the topological susceptibility for QCD instantons. We stress that an explicit calculation of this quantity for colored gravitational instantons is needed, though this is beyond the scope of the present work.

---

[2]Non-linearities in a two-axion potential may lead to resonant energy transfer between the two particles as recently studied in the context of string axiverse models [50]. In the companion-axion model this is unlikely to happen, since it would require $\epsilon \lesssim 0.2$ and $m_2 \approx m_1$, at the same time.

Since the time dependence has been factored out in Eq.(5), we can decouple the system of equations (4) by working in the mass basis $a_{1,2}$:

$$\partial_t^2 a_1 + \frac{3}{2t}\partial_t a_1 + m_1^2(T)a_1 = 0\,,$$
$$\partial_t^2 a_2 + \frac{3}{2t}\partial_t a_2 + \kappa\epsilon^2 m_1^2(T)a_2 = 0\,. \tag{7}$$

We define $t_1$ $(T_1)$ as the time (temperature) at which the zero-mode of heavier axion $a_1$ starts oscillating due to the presence of the mass term in Eq.(7); and $t_2$ $(T_2)$ for the lighter axion. This happens when $m_1(T_1) = 3H(T_1)$ and $m_2(T_2) = 3H(T_2)$ respectively, where

$$T_1 = \left(\frac{m_1 M_{\mathrm{P}}\sqrt{90}}{24\pi^2\sqrt{g_*^1}}\right)^{\frac{2}{n+4}} \widetilde{T}^{\frac{n}{n+4}}\,, \tag{8}$$

with $g_*^1$ the relativistic degrees of freedom at $T_1$ and $M_{\mathrm{Pl}} \simeq 1.2 \times 10^{19}$ GeV. This is a general definition valid for any $\epsilon$, but in the hierarchical regime of our model (when $\epsilon \ll 1$) the temperature at the onset of oscillations for the lighter axion is smaller by a factor $T_2/T_1 \sim (\kappa\epsilon^2)^{1/(n+4)}$. When $m_1 \sim$ μeV, the first oscillations starts at temperatures $T_1 \sim \mathcal{O}(\mathrm{GeV})$, as in the standard single-axion case.

The solutions to Eqs.(7) can be readily obtained via a Wentzel–Kramers–Brillouin (WKB) approximation and the energy density stored in the oscillations of the axions follows from an average over multiple oscillations. Assuming comoving entropy-density conservation, the axion energy densities are:

$$\rho_i|_{\mathrm{today}} = m_i(T_i)m_i\langle a_i^2\rangle\left(\frac{T_0}{T_i}\right)^3\frac{g_{*s}^0}{g_{*s}^i}\,, \tag{9}$$

where $T_0 = 2.7$ K, and $g_{*s}^i = g_{*s}(T_i)$ are the entropy degrees of freedom.

The total relic abundance is the sum of the two components $\Omega_a = \Omega_1 + \Omega_2$. We find for the hierarchical regime ($\epsilon \ll 1$),

$$\Omega_a h^2 = \Omega_1 h^2\left(1 + \frac{\theta_2^2}{\theta_1^2}\frac{g_{*s}^1}{g_{*s}^2}\kappa^{\frac{n+2}{2(n+4)}}\epsilon^{-\frac{n+6}{n+4}}\right) \quad (\epsilon \ll 1). \tag{10}$$

Here the misalignment angles are defined as $\theta_1 = \langle a_1(t_1)\rangle/f_a$ and $\theta_2 = \langle a_2(t_2)\rangle\epsilon/f_a$. When the PQ breaking scales are hierarchical, the lighter axion dominates the relic abundance by a factor $\sim \epsilon^{-1.19}$ (for $n = 6.68$)[3], unless $\theta_2 \ll \theta_1$, which could occur in Cases I and II.

In the strong mixing regime ($\epsilon \lesssim 1$) the thermal mass of the lighter axion is $m_2(T) \sim \sqrt{\kappa}m_1(T)$, and we can proceed similarly in the estimation of the relic abundance to find,

$$\Omega_a h^2 = \Omega_1 h^2\left(1 + \frac{\theta_2^2}{\theta_1^2}\kappa^{\frac{n+2}{2(n+4)}}\right) \quad (\epsilon \lesssim 1), \tag{11}$$

where we have taken $g_{*s}^1 = g_{*s}^2$. The $\kappa^{0.41}$ dependence in Eq.(10) and (11) comes from the mass dependence in Eq.(9), and competes with the effect of $\epsilon$. So if the decay constants $f_a$, $f_a'$ are close to each other and the misalignment angles are of the same order, $a_1$ slightly dominates due to its larger mass. However in all other cases, $a_2$ dominates.

We can now use the results presented above to derive preferred regions of parameter space for which the companion axions constitute all of the DM. This is shown in Fig. 1. In the left-hand panel we show the result for Case I, when both PQ symmetries are broken before the end of inflation. Since we have freedom to choose any value of $\theta_{1,2}$, we show a "natural" window where neither angle has to be tuned to within $\delta = 0.1$ of the angles 0 or $\pi$. This is perhaps the most novel scenario for the companion-axion model. We have much more freedom to match the DM abundance here, because the two axions can be traded off for one another. This feature of the

---

[3]The entropy degrees of freedom should not change substantially from $T_1$ to $T_2$

model is shown in Fig. 2 where, rather than a single value of $\theta$ matching the correct DM abundance for a given set of model parameters, we have an arc of values in the $(\theta_1, \theta_2)$ plane.

The second panel of Fig. 1 shows Case II as the colored area, whereas Case III only appears as a thin line within this band since it is effectively a special case of Case II when $\theta_2 = \pi/\sqrt{3}$. As we can see, the axion abundance is generally always dominated by the lighter axion $\Omega_2 > 0.5$, except when $\epsilon$ is close to 1, or if $\theta_2$ needs to be tuned towards $\pi$ to avoid over-production. The preferred window of axion masses is similar to the usual calculation for the heavier axion: $m_1 \sim 10^{-6}$–$10^{-4}$ eV, but for $m_2$ we predict a substantially lighter window: $m_2 \sim 10^{-8}$–$10^{-6}$ eV . Both of these regions should be accessible with future haloscopes [51–68]. For now we have only shown the region already excluded by ADMX, assuming the KSVZ-like photon couplings derived in Ref. [15].

## 4  Isocurvature bounds

In our pre-inflationary cases, the massless axion fields will undergo large amplitude quantum fluctuations during inflation. The energy density of these fluctuations is negligible compared to the dominant energy density carried by the inflaton field and hence do not contribute to the perturbations of the total energy density. However, they contribute non-negligibly to the perturbations of the ratio of axion number density to entropy, giving rise to the so-called entropy, or *isocurvature*, perturbations [69–72]. As the axions would survive past recombination in the form of DM, such perturbations contribute to the temperature and polarisation fluctuations in the cosmic microwave background radiation and are uncorrelated with the inflaton (adiabatic) fluctuations. Assuming that each of the companion axion fluctuations are also uncorrelated, we can compute the perturbation power spectrum at the pivot scale $k_{\text{low}} \approx 0.002\,\text{Mpc}^{-1}$ as:

$$\Delta_{a_1}^2 = \Delta_{a_2}^2 \epsilon^{-2} \frac{\theta_2^2}{\theta_1^2} \simeq \frac{H_I^2}{\pi^2 f_a^2 \theta_1^2} , \tag{12}$$

where $H_I$ is the Hubble expansion rate during inflation. Since the total primordial power spectrum is dominated by the adiabatic fluctuations, its amplitude at the pivot scale can be approximated as $A_s \simeq H_I^2/(\pi^2 M_{\text{Pl}}^2 \varepsilon) \approx 2 \times 10^{-9}$, where $\varepsilon$ is the inflation 'slow roll' parameter. Hence, the isocurvature power spectrum ratio, for Case I, can be written as:

$$\beta = \frac{\Delta_{a_1}^2}{A_s}\left(1 + \epsilon^{-2}\frac{\theta_2^2}{\theta_1^2}\right) , \tag{13}$$

leading to,

$$H_I \lesssim \frac{\sqrt{\beta_\star A_s}\,\pi f_a \theta_1}{(1 + \epsilon^{-2}\theta_2^2/\theta_1^2)^{1/2}}, \qquad (\text{Case I}). \tag{14}$$

We have written this in terms of the constraint from Planck: $\beta < \beta_\star = 0.011$ (95 % C.L.) [73] at $k_{\text{low}}$. In Fig. 2 we showed how this constraint depends upon the two $\theta$ angles, for a particular illustrative choice of $f_a$ and $\epsilon$. We see that when $\theta_1$ must be tuned very close to 0, we are only allowed a rather low scale of inflation.

When $\epsilon \ll 1$ however, it might be more prudent to consider Case II, where the heavier axion lands in the post-inflationary scenario. In this case, the isocurvature bounds on the companion axion model can be simply drawn from the literature on the single axion case, with $f_a \rightarrow f_a' = f_a/\epsilon$.

## 5  Domain Walls and Gravitational Waves

Axion domain walls emerge because of a residual discrete $Z_N \times Z_{N'}$ shift symmetry that the potential Eq.(1) enjoys in the absence of the second cos-term. The axion expectation values $\langle a \rangle$ and $\langle a' \rangle$ break this symmetry spontaneously, resulting in the formation of a cosmological domain

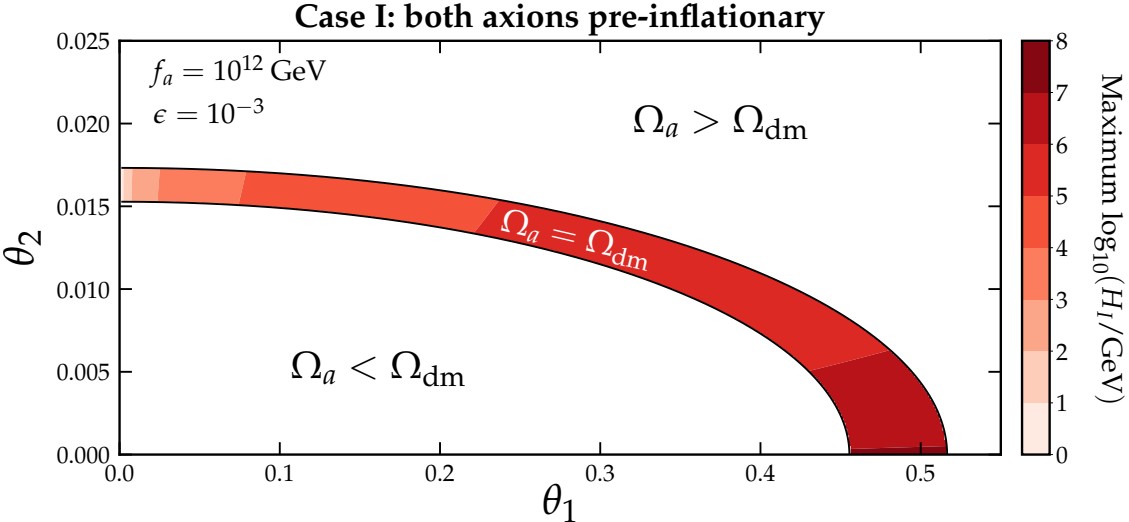

Figure 2: Illustration of the freedom given in our Case I, when both axions are pre-inflationary. Since we have the freedom to choose two initial angles, we can trade one axion off with the other while satisfying the correct DM abundance. This is shown by the arc in the $(\theta_1, \theta_2)$ plane. Additionally we also color the region by the maximum value of $H_I$ allowed by Planck isocurvature bounds, for that particular choice of parameters. We emphasize that the white regions plotted here are *not* constrained from comprising the correct DM fraction, since the position of the arc depends on the particular values of $f_a$ and $\epsilon$ chosen.

wall network that can overclose the Universe, unless made unstable somehow. This is known in the literature as the *domain wall problem* [74–76]. In the companion axion model, however, the additional instantons that result in the second cos-term in Eq.(1) explicitly break the residual discrete symmetry of the first cos-term, and *vice versa*. Hence the discrete degeneracy of axion vacuum states is lifted and the energy difference,

$$V_{\text{bias}} \sim \kappa K, \tag{15}$$

acts like a bias-term to drive the annihilation of the domain walls (see e.g. [77–85] for many alternative realizations of this effect). We can state, therefore, that the domain wall problem is automatically solved in the companion-axion model.

Explicit solutions for $Z_N \times Z_{N'}$ domain walls are quite involved so we proceed by making an order of magnitude estimation. A network of axionic domain walls starts to form during the QCD phase transition via the Kibble mechanism [86, 87]. Axion strings, which form from the spontaneous symmetry breaking of the global $U(1)_{\text{PQ}} \times U(1)'_{\text{PQ}}$ symmetry[4], are attached to each domain wall junction. The domain walls, like the strings, are expected to enter a *scaling regime*, where the network tends to contain $\mathcal{O}(1)$ walls per Hubble volume [89]. This means that the typical distance between two neighboring walls is given by the Hubble radius, $r_H \sim t$. The energy density of domain walls is $\rho_w \sim \sigma/t$, with $\sigma$ the surface tension, and so decays much slower than matter or radiation. In the companion-axion model, two sets of domain walls appear[5], of width given by the Compton wavelength of the corresponding axion, $\delta_{1,2} \sim 1/m_{1,2}$. The energy barrier

---

[4]One should keep in mind that since the companion axion model involves two PQ scalar fields, the symmetry non-restoration scenario can in principle be realized for a certain region of parameter space [88]. The domain wall problem is then resolved simply because there are no transitions and hence the domain wall production is heavily suppressed.

[5]More complicated hybrid solutions are also possible [90].

that separates discrete vacua $V_0$ can be used to estimate the surface tensions $\sigma_i$ as,

$$\sigma_i \sim V_0 \delta_i, \quad V_0 \sim K(1+\kappa), \tag{16}$$

respectively, underlining how walls associated with the lighter axion are wider and more energetic. The energy density difference $V_{\text{bias}}$ acts as a volume pressure $p_V$ on the walls, meaning that a domain wall of size $r \sim t$ gets annihilated when this pressure starts to dominate over the wall surface tension, $p_T \sim \sigma/t$. Hence, we can estimate the wall annihilation time when this happens, $t_{\text{ann}} \sim \sigma/V_{\text{bias}}$. In the radiation dominated era, it corresponds to the temperature,

$$T_{\text{ann}} \sim 13.5 \text{ MeV} \left( \frac{m_i}{10^{-12} \text{ eV}} \right)^{1/2} \left( \frac{11\kappa}{1+\kappa} \right)^{1/2} \left( \frac{10}{g_*} \right)^{1/4} . \tag{17}$$

Although the domain walls annihilate before Big Bang nucleosynthesis ($T_{\text{BBN}} \sim 1$ MeV), for consistency we also require $T_{\text{ann}}(m_{1,2}) \lesssim T_{1,2}$. This condition is violated by axion masses $m_i \gtrsim 10^{-9}$ eV, with the interpretation that the bias potential is so large that the walls cannot even form.

Additionally, describing the field in terms of domain walls is only meaningful if their widths do not exceed the horizon size, $m_i \gtrsim H(T)$, which is satisfied for $m_i \gtrsim 10^{-10}$ eV. Hence the dynamics of domain walls is relevant only for a narrow range of masses $10^{-10}$ eV $\lesssim m_i \lesssim 10^{-9}$ eV. This also implies that for the majority of the parameter space of interest, we can state that the non-relativistic production of axions from the wall network does not contribute much to the DM abundance, and the estimation of the misalignment production discussed earlier should not change substantially.

In the cases where domain walls do form, violent collisions from their decay will produce strong metric perturbations which can result in GWs [91]. Detailed numerical simulations of GW production from wall decay have been carried out in Refs. [92–94] (see also the review [95]). The GW power spectrum was observed to grow as $\sim k^3$ up to a peak comoving wavenumber $k_{\text{peak}}$ (as expected by correlation arguments), above which it falls off as $\sim k^{-1}$, with a cutoff set by the wall thickness. The peak frequency can be estimated as $f_{\text{peak}} = k_{\text{peak}}/2\pi R(t)$ giving,

$$f_{\text{peak}} \sim 1.1 \times 10^{-8} \text{ Hz} \left( \frac{m_i}{10^{-10} \text{ eV}} \right)^{1/2} \left( \frac{11\kappa}{1+\kappa} \right)^{1/2} , \tag{18}$$

where we have taken $g_*(T_{\text{ann}}) = g_{*s}(T_{\text{ann}}) = 10$. The relic density at the peak is,

$$\left( \Omega_{\text{gw}} h^2 \right)_{\text{peak}} \sim 3 \times 10^{-10} \left( \frac{10^{-10} \text{ eV}}{m_i} \right)^4 \left( \frac{(1+\kappa)^2}{12.1\kappa} \right)^2 . \tag{19}$$

Figure 3 shows the predicted signals of GWs expressed in terms of the relic density as a function of frequency. The bands for each mass span the uncertainty on the parameter $\kappa$ [14]. We also show power-law integrated exclusion curves [97, 98] for previous pulsar timing array searches for a GW background (NANOGrav [99–102], EPTA [103–105] and PPTA [106, 107]) as well as the future sensitivities of LISA [108] and SKA [109–111]. We make use of the power-law integrated curves presented in Ref. [96] with the exception of the sensitivity to GWs with astrometric data, which is taken from Ref. [112].

We can see the mass range, $10^{-10}$ eV $\lesssim m_i \lesssim 10^{-9}$ eV, can be probed by searching for a stochastic gravitational background with $>$nHz frequencies, as shown in Fig. 1. While the signal remains just out of reach in current pulsar timing arrays, masses up to around $5 \times 10^{-9}$ eV should be within reach of SKA. Interestingly, the largest example we predict here is close to the strong signal of a stochastic background reported recently by NANOGrav [113]—though interpreting this signal as due to GWs is premature at this stage. Merely for context, we show the rough expected range of frequencies and amplitudes that the signal would correspond to if it were a GW background, following Ref. [114].

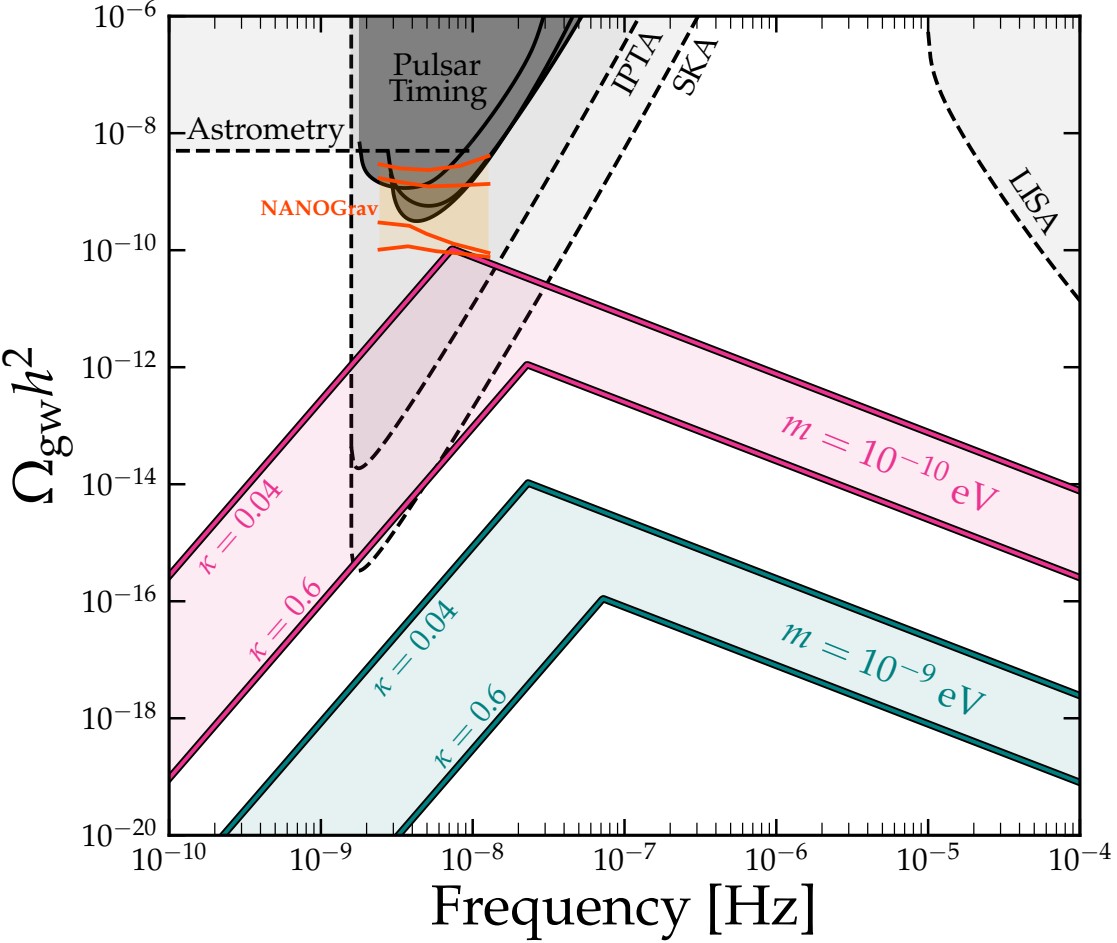

Figure 3: Relic abundance of GWs from wall decay in our model. We show constraints from pulsar timing arrays (NANOGrav-11 yr, EPTA and PPTA), the NANOGrav-12.5 yr hint (shown at $1/2\sigma$) and the sensitivities of SKA, LISA, and an astrometry-based search for a stochastic GW background. GW curves are power-law integrated sensitivities taken from Ref. [96].

## 6 Primordial Black Holes

Another potential consequence of the domain wall network are black holes [115], which could be formed from the collapse of closed domains containing a 'false' vacuum. Closed domains will start shrinking once their sizes approach the Hubble horizon, $r \sim 1/H$, with an energy comprised of the wall tension (surface effect) and the interior false vacuum energy (volume effect):

$$M_i = 4\pi r_i^2 \sigma_i + \frac{4\pi}{3} r_i^3 V_{\text{bias}} \,. \tag{20}$$

The domain will collapse into a black hole if its size is less than the Schwarzschild radius corresponding to the mass in Eq.(20): $r_i = 2M_i/M_{\text{P}}^2$ [115]. For the range of axion masses relevant for domain wall formation, the bias term dominates Eq.(20) and we can simply estimate the masses of black holes produced as,

$$M_{\text{PBH}} \sim \frac{\sqrt{3}}{4\sqrt{2}} \frac{M_P^3}{(\pi \kappa K)^{1/2}} \sim 150 \, M_\odot \left( \frac{\kappa}{0.1} \right)^{-1/2} \,. \tag{21}$$

The temperature of the Universe when this collapse occurs is,

$$T_{\text{coll}} \sim 25 \text{ MeV} \left(\frac{\kappa}{0.1}\right)^{1/4} \left(\frac{g_*}{10}\right)^{-1/4} . \tag{22}$$

The value of the latter, within our rough estimation, can be as small as $\sim 20$ MeV, considering our uncertainty on $\kappa$.

PBHs generated by the collapse of axion domain walls were also considered in Ref. [29, 30]. Although our estimate is similar to these earlier studies, there are some qualitative differences. For example, in order to preserve the single QCD axion solution to the strong CP problem, Ref. [30] considered a very small bias term (equivalent to $\kappa \sim 10^{-12}$ in our model) and the formation of PBHs relied on the longevity of a $N_{\text{DW}} > 1$ domain wall network. Additionally, the shorter lifetime of the wall network meant that fewer domains survive down to the collapse temperatures.

To estimate the survival probability, Ref. [30] used a power-law fit to numerical simulations [37] of a simple domain wall network. However, because of the complexity of the domain wall network and the large bias potential, use of these results for the companion-axion model would be unreliable. Instead, we make a very crude analytical estimate as follows. The process under consideration can be treated as the decay of the false vacuum with mean lifetime $\sim t_{\text{ann}}$ (17). The fraction of domains that survive decay by the time $t_{\text{coll}}$ (22) is then,

$$p_{\text{coll}} \sim e^{-(T_{\text{ann}}/T_{\text{coll}})^2} \sim 10^{-22} - 10^{-9}, \tag{23}$$

where the range corresponds to $\kappa \in [0.04, 0.6]$, while fixing the axion mass $m_i = 10^{-10}$ eV. For larger $m_i$ the survival probability is essentially zero, because of the mass dependence in Eq. (17).

Given Eq.(23) we can compute the present-day PBH energy density,

$$\rho_{\text{PBH}} \sim \frac{p_{\text{coll}}}{r_i^3} M_{\text{PBH}} \left(\frac{T_0}{T_{\text{coll}}}\right)^3 \sim p_{\text{coll}} \frac{M_{\text{P}}^6}{M_{\text{PBH}}^2} \left(\frac{T_0}{T_{\text{coll}}}\right)^3 , \tag{24}$$

and therefore the fraction of the DM density they constitute:

$$f_{\text{PBH}} = \frac{\rho_{\text{PBH}}}{\rho_{\text{dm}}} \simeq 34.9 \, p_{\text{coll}} \frac{M_{\text{P}}^4}{H_0^2 M_{\text{PBH}}^2} \left(\frac{T_0}{T_{\text{coll}}}\right)^3 . \tag{25}$$

This estimate inherits large uncertainties from Eq.(23), such that the PBH dark matter fraction ranges from $\mathcal{O}\left(10^{-13}\right)$ to $\mathcal{O}(1)$ for the axion mass $m_i \sim 10^{-10}$ eV. For heavier axions than this, PBH abundance is negligibly small.

Though confined to a narrow area of the theory's parameter space, it is still intriguing that the model provides a mechanism for the formation of a subdominant population of LIGO-sized PBHs [116]. The DM fraction in black holes of size $\sim 100 M_\odot$ has only relatively weak constraints [117–119], but this is a mass range that will receive significant interest in the coming years via more sensitive GW observations.

# 7 Conclusions

It was shown recently [14] that the strong CP problem cannot be resolved in a single-axion model once the effects of colored gravitational instantons are taken into account. This necessitates the extension of the PQ mechanism via a second axion that cancels off the additional unwanted CP-violation [14, 15]. Here, we have investigated the cosmological viability and implications of this new QCD axion model. Specifically, we have calculated the abundance of axions produced via the misalignment mechanism. Instead of the standard two scenarios (pre/post-inflation), in the companion-axion model we have three scenarios, depending on the sizes of the two axions' symmetry breaking scales, and the scale of inflation. In general, the lighter axion dominates the DM abundance, unless the symmetry breaking scales are finely tuned to the same value.

Notably, in the purely pre-inflationary scenario (Case I) we find that there is considerable

freedom to satisfy the correct quantity of DM in the Universe, since one can always trade the abundance of one axion for another by tuning the initial misalignment angles accordingly (see Fig. 2). Nevertheless, we can identify a preferred window which does not require fine-tuning of the two misalignment angles, as is shown in the left-hand panel of Fig. 1. Generally, for all scenarios we arrive at typical preferred axion mass ranges of $m_1 \sim 10^{-6}$–$10^{-4}$ eV for the heavier axion, and $m_2 \sim 10^{-8}$–$10^{-6}$ eV for the lighter one. Both of these windows are within reach of future experiments—see Ref. [15] for further discussion.

Our two post-inflationary scenarios (Cases II and III), lead to the most dramatic cosmological implications. The most interesting of these is the potential formation of domain walls. In the companion-axion model the second axion acts like a bias term, rendering the domain walls unstable and resolving the domain wall problem automatically. For the small parts of parameter space where this process takes place (for $m_i \sim 10^{-10} - 10^{-9}$ eV), we predict additional signals of the companion axion model, such as the generation of 10 nHz GWs, as well as $\sim 100\, M_\odot$ LIGO-sized primordial black holes. Therefore it may even be possible for a positive signal of two QCD axions in a laboratory experiment to be combined with one of these aforementioned gravitational signals, so as to eventually study the free parameters of the companion-axion model much more precisely.

The figures from this article can be reproduced using the code available at:
$$\text{https://github.com/cajohare/CompAxion,}$$
whereas the data for haloscope limits is compiled at Ref [120].

## Acknowledgements

This paper was partially written on Gadigal lands. This work made use of NUMPY [121], SCIPY [122], and MATPLOTLIB [123].

## Funding

The work of AK was partially supported by the Australian Research Council through the Discovery Project grant DP210101636 and by the Shota Rustaveli National Science Foundation of Georgia (SRNSFG) through the grant DI-18-335.

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
