# Peer review of "Cosmology of the companion-axion model: dark matter, gravitational waves, and primordial black holes"

_SciPost Physics, doi:SciPost Phys. 18, 175 (2025)_

## Round 1 · Referee Report · Anonymous (Referee 2) · 2025-1-30

Report

In this paper, the authors studied the cosmology of two QCD axion fields in the presence of two potentials, one from the usual QCD instanton and the other from what they call colored gravitational instantons. They calculated the dark matter abundance, isocurvature perturbations, and gravitational waves from domain wall collapse in this model.

I am not particularly convinced by their starting point, Eq. (1). Their claim, based on their previous paper Ref. [14], is that gravity introduces an additional potential to the axion whose size is not suppressed by the Planck scale and can be as large as 0.1 times the standard QCD potential. This is a strong claim, and as far as I know, this is far from being widely accepted. Therefore, I think the authors should, at least, add an explanation as to why they believe that gravity effects, which I would normally expect to be suppressed by the Planck scale, have such a huge contribution so that readers can notice this potential caveat. I know that this was the topic of their previous paper, but it makes sense to me to add an explanation again. Related to this point, the authors may address how their result is affected by the size of $\kappa$, as this parameter controls the gravitational contribution.

Putting this point aside, I can view this paper as a study of the cosmology of a two-axion(-like particle) model. However, there is previous literature studying this topic, and the analysis in this paper does not seem particularly new in this respect. Given my suspicion of their starting point, I therefore cannot recommend this paper for publication.

Recommendation

Reject

---

## Round 1 · Referee Report · Anonymous (Referee 3) · 2025-2-12

Report

This is the referee report to the manuscript entitled “cosmology of the companion-axion model: dark matter, gravitational waves, and primordial black holes” by Zhe Chen, Archil Kobakhidze, Ciaran O’Hare, Zachary S. C. Picker, and Giovanni Pierobon.

In this manuscript, the authors study cosmological aspects of the companion-axion model, an extention of the original Peccei-Quinn model, which is motivated by the problem with the original solution to the strong CP problem. Specifically, the authors focus on the cosmological evolution of two axions predicted in this model.

First, the authors have carefully derived the relic dark matter abundance in this two axion model and shown the viable parameter region in terms of the corresponding two axion decay constants, which is summarized in Fig. 1. It predicts the relatively heavy axion and the ligher axion. The accessibility by the future cavity experiments is also shown. Then, the authors have derived the isocurvature bound in their model. The bound on the inflationary Hubble scale in terms of the two initial misalignment angles is calculated and summarized in Fig. 2.

Next, the authors focus on the domain wall formation in their model. In particular, the domain wall annihilation with the potential bias term and the emission of gravitational waves are analysed. The authors estimated the peak frequency and the relic density parameter of the predicted stochastic gravitational wave banckground, which is summerized in Fig. 3. Finally, the authors study the primordial black hole formation from domain walls in this model. The mass and the abundance of PBHs are roughly estimated, and as a result ~100 solar mass PBH is predicted.

On the whole, the discussion is clear and the results have enough scientific impact. I think however the manuscript needs some minor revisions as listed below.

  • Although the reader can refer the previous works[14,15], the motivation of the companion axion model should be written in more detail, such as the brief introduction of the charged gravitational instantons, extra CP-violation, the problem of the original PQ solution, the physical meaning of the kappa parameter (in eq (1)) and reason for its viable range, 0.04 - 0.6 (because it is important for the domain wall annihilation).

  • In the r.h.s. of eq. (1), I think the second term in the cosine function should be a’ (not a).

  • The authors should comment on the parameters theta and theta_g in eq. (1), especially, the viable range of these parameters.

Once the above issues are addressed, I think this manuscript satisfies the acceptance criteria and desrves the publication in SciPost Physics.

Recommendation

Ask for minor revision

---

## Round 2 · Referee Report · Anonymous (Referee 2) · 2025-4-29

Report

The authors replied to my previous comments and implemented the changes accordingly. Even though I still personally do not buy their argument on the size of the additional gravitational effect, I can view it as a study on a two-field axion model. It is a fair point that this paper appeared on arXiv rather early, and given that the authors have added a paragraph on other references and emphasized their difference, I may now recommend it for publication.

Recommendation

Publish (meets expectations and criteria for this Journal)

---

## Round 2 · Referee Report · Anonymous (Referee 3) · 2025-5-16

Report

The authors have fully addressed the issues raised, and thus I recommend the paper for publication in SciPost Physics.

Recommendation

Publish (meets expectations and criteria for this Journal)

---

## Round 2 · Author Response

Report #3
"This is the referee report to the manuscript entitled “cosmology of the companion-axion model: dark matter, gravitational waves, and primordial black holes” by Zhe Chen, Archil Kobakhidze, Ciaran O’Hare, Zachary S. C. Picker, and Giovanni Pierobon.
In this manuscript, the authors study cosmological aspects of the companion-axion model, an extention of the original Peccei-Quinn model, which is motivated by the problem with the original solution to the strong CP problem. Specifically, the authors focus on the cosmological evolution of two axions predicted in this model.
First, the authors have carefully derived the relic dark matter abundance in this two axion model and shown the viable parameter region in terms of the corresponding two axion decay constants, which is summarized in Fig. 1. It predicts the relatively heavy axion and the ligher axion. The accessibility by the future cavity experiments is also shown. Then, the authors have derived the isocurvature bound in their model. The bound on the inflationary Hubble scale in terms of the two initial misalignment angles is calculated and summarized in Fig. 2.
Next, the authors focus on the domain wall formation in their model. In particular, the domain wall annihilation with the potential bias term and the emission of gravitational waves are analysed. The authors estimated the peak frequency and the relic density parameter of the predicted stochastic gravitational wave banckground, which is summerized in Fig. 3.
Finally, the authors study the primordial black hole formation from domain walls in this model. The mass and the abundance of PBHs are roughly estimated, and as a result ~100 solar mass PBH is predicted.
On the whole, the discussion is clear and the results have enough scientific impact. I think however the manuscript needs some minor revisions as listed below."

We thank the referee for their thorough understanding of our work, and are pleased to see that they consider the work worthy of publication in SciPost.

"- Although the reader can refer the previous works[14,15], the motivation of the companion axion model should be written in more detail, such as the brief introduction of the charged gravitational instantons, extra CP-violation, the problem of the original PQ solution, the physical meaning of the kappa parameter (in eq (1)) and reason for its viable range, 0.04 - 0.6 (because it is important for the domain wall annihilation)."

The introduction has been significantly expanded. The second paragraph now contains a more detailed review of the gravitational instantons, the resulting axion potential from their inclusion, how this spoils the strong-CP solution, and some clarifying notes on the kappa parameter.
We have also added a paragraph beginning with “Two-axion theories…” which summarizes some of the existing literature on multiple axion scenarios.

"- In the r.h.s. of eq. (1), I think the second term in the cosine function should be a’ (not a)."

This has been fixed.

"- The authors should comment on the parameters theta and theta_g in eq. (1), especially, the viable range of these parameters.
This is now included in the paragraph following the new eq.1 in the introduction.
Once the above issues are addressed, I think this manuscript satisfies the acceptance criteria and desrves the publication in SciPost Physics."

We again thank the referee for their useful advice in preparing this manuscript, and for their approval of its publication in SciPost.
* * *
Report #2

"In this paper, the authors studied the cosmology of two QCD axion fields in the presence of two potentials, one from the usual QCD instanton and the other from what they call colored gravitational instantons. They calculated the dark matter abundance, isocurvature perturbations, and gravitational waves from domain wall collapse in this model."

We thank the referee for their thorough and insightful questions. They have greatly helped us clarify our points in the manuscript, and we hope that our responses here and the changes made therein have adequately addressed their concerns.

"I am not particularly convinced by their starting point, Eq. (1). Their claim, based on their previous paper Ref. [14], is that gravity introduces an additional potential to the axion whose size is not suppressed by the Planck scale and can be as large as 0.1 times the standard QCD potential. This is a strong claim, and as far as I know, this is far from being widely accepted. Therefore, I think the authors should, at least, add an explanation as to why they believe that gravity effects, which I would normally expect to be suppressed by the Planck scale, have such a huge contribution so that readers can notice this potential caveat. I know that this was the topic of their previous paper, but it makes sense to me to add an explanation again. Related to this point, the authors may address how their result is affected by the size of κ, as this parameter controls the gravitational contribution."

The introduction has been significantly expanded. The second paragraph now contains a more detailed review of the gravitational instantons, the resulting axion potential from their inclusion, how this spoils the strong-CP solution, and some clarifying notes on the kappa parameter. Throughout the paper, results are given for the range of kappa estimated from our previous works, quantitatively showing its contribution.

The question about the impact of the Planck scale is an excellent question, and we have added some discussion of it to the introduction of our manuscript as well—we thank the referee for helping us consider this more carefully in our writing. The heuristic reason that the effects are closer to the QCD scale than the gravity scale is because this is not a pure gravity instanton solution (which has been considered before, but does not affect the QCD CP-problem). Rather, this is a mixed QCD/gravity instanton known as a color-charged Eguchi-Hanson instanton. It is from the colored part of these instantons that the scale becomes closer to the QCD scale. For a detailed mathematical explanation, we point the referee to arXiv:9207208, which we have now additionally cited in this manuscript. This work carefully steps through the charged Eguchi-Hanson instanton, showing that it can, and does, affect the PQ axion well above the Planck scale (see the discussion following eq.9, for instance). This paper considered electrically charged EH instantons, but this formalism was extended to color-charged ones in the work of Kobakhidze and Chen which then formed the basis for this paper. These previous papers that are the foundation for this work were both published in reputable journals and remain well cited in this niche, and we do not agree that the results of these works are “far from being widely accepted.”

"Putting this point aside, I can view this paper as a study of the cosmology of a two-axion(-like particle) model. However, there is previous literature studying this topic, and the analysis in this paper does not seem particularly new in this respect. Given my suspicion of their starting point, I therefore cannot recommend this paper for publication."

We have added a paragraph beginning with “Two-axion theories…” which summarizes some of the existing literature on multiple axion scenarios. Studies which existed prior to this paper had different motivation (mainly from axiverse scenarios) and so the resulting axion models differed significantly. In particular, our companion axion scenario requires two coupled axions in order to solve the strong-CP problem. This specific coupling leads to different phenomenology which we describe in this paper. As far as we know, this had never been investigated before and the results—particularly related to the dark matter abundance—were unique and potentially crucial for future axion searches. Another important result here was the non-existence of the domain wall problem, a serious theoretical hurdle that is naturally solved in our model.

In fact, as we explained in the submission, this preprint is from 2021, and for various personal reasons the authors did not seek to publish it at the time. Since then, the preprint has accumulated a steady stream of citations (actually, more than either of our other published companion axion papers), indicating that it has results which were both novel and useful for many working in this area since then. Thus some of this indicated ‘previous’ literature may in fact be following on from our work here. If there are additional relevant papers that predate this work, and which we have not addressed in this revision, we would gladly discuss and cite them.
* * *

---

## Round 2 · List of Changes

The introduction to the manuscript is significantly expanded. It now includes a review of previous literature relating to multiple axion scenarios, and a significantly more detailed introduction to the charged Eguchi-Hanson instantons and the motivation for this work.

The small typos and revisions indicated by Report #3 have been fixed.

---

## Editorial Decision

published